# Genome-Wide Association Studies Reveal That the Abietane Diterpene Isopimaric Acid Promotes Rice Growth through Inhibition of Defense Pathways

**DOI:** 10.3390/ijms25179161

**Published:** 2024-08-23

**Authors:** Xiaomeng Luo, Liping Bai, Jiaqi Huang, Luying Peng, Juan Hua, Shihong Luo

**Affiliations:** College of Bioscience and Biotechnology, Shenyang Agricultural University, Shenyang 110866, China

**Keywords:** rice, isopimaric acid, Abietane diterpenes, plant growth regulators, GWAS, resistance genes

## Abstract

Plants are an important source for the discovery of novel natural growth regulators. We used activity screening to demonstrate that treatment of Nipponbare seeds with 25 μg/mL isopimaric acid significantly increased the resulting shoot length, root length, and shoot weight of rice seedlings by 11.37 ± 5.05%, 12.96 ± 7.63%, and 27.98 ± 10.88% and that it has a higher activity than Gibberellin A3 (GA_3_) at the same concentration. A total of 213 inbred lines of different rice lineages were screened, and we found that isopimaric acid had different growth promotional activities on rice seedlings of different varieties. After induction with 25 μg/mL isopimaric acid, 15.02% of the rice varieties tested showed increased growth, while 15.96% of the varieties showed decreased growth; the growth of the remaining 69.02% did not show any significant change from the control. In the rice varieties showing an increase in growth, the shoot length and shoot weight significantly increased, accounting for 21.88% and 31.25%. The root length and weight significantly increased, accounting for 6.25% and 3.13%. Using genome-wide association studies (GWASs), linkage disequilibrium block, and gene haplotype significance analysis, we identified single nucleotide polymorphism (SNP) signals that were significantly associated with the length and weight of shoots on chromosomes 2 and 8, respectively. After that, we obtained 17 candidate genes related to the length of shoots and 4 candidate genes related to the weight of shoots. Finally, from the gene annotation data and gene tissue-specific expression; two genes related to this isopimaric acid regulation phenotype were identified as *OsASC1* (*LOC_Os02g37080*) on chromosome 2 and *OsBUD13* (*LOC_Os08g08080*) on chromosome 8. Subcellular localization analysis indicated that OsASC1 was expressed in the plasma membrane and the nuclear membrane, while OsBUD13 was expressed in the nucleus. Further RT-qPCR analysis showed that the relative expression levels of the resistance gene *OsASC1* and the antibody protein gene *OsBUD13* decreased significantly following treatment with 25 μg/mL isopimaric acid. These results suggest that isopimaric acid may inhibit defense pathways in order to promote the growth of rice seedlings.

## 1. Introduction

Plant growth regulators are naturally or artificially synthesized compounds that are physiologically similar to endogenous phytohormones [1]. Those metabolites regulate life processes in plants, including flowering, dormancy, growth, and germination [2]. Moreover, growth regulators have been widely used in field crops, fruiting trees, vegetables, flowers, and other crops, leading to stable yields, superior product quality, and enhanced crop resistance to diseases, all of which are important for the promotion of agricultural production [3].

Depending on their function, plant growth regulators can be divided into plant growth promoters, plant growth retardants, or plant growth inhibitors. Of these, plant growth promoters are the most versatile and commonly utilized in agriculture [4]. These compounds facilitate cell division, differentiation, and elongation, as well as stimulating the growth of vegetative organs and the development of reproductive organs in plants [5]. For example, the auxin analog 1-naphthylacetic acid (NAA) can stimulate plant cell division and growth, induce random root formation, increase fruit yield, and enhance plant metabolism and photosynthesis, thereby accelerating growth and development [6,7]. Diethylaminoethylhexanoate (DA−6) has been found to enhance plant physiological processes such as the activities of oxidase and nitrate reductase, leading to improved carbon and nitrogen metabolism, enhanced water and fertilizer absorption, increased dry matter accumulation, and enhanced plant quality. Additionally, 6-benzylaminopurine (6-BA) can stimulate cell division, induce flower bud differentiation, increase fruit setting rate, and promote fruit growth [8,9].

Most currently used commercial plant growth regulators are artificially synthesized and directly related to plant endogenous hormones. Other natural active substances able to regulate growth in plants are mostly derived from fungi. One example is cremenolide, a macrolide found in *Trichoderma cremeum*, which is able to promote the growth of tomato seedlings [10]. Similarly, two non-volatile stereoisomers, *3S,4R*-acetylbutanediol and *3R,4R*-acetylbutanediol from *Bacillus velezensis* WRN031, can promote the growth of maize and rice seedling roots [11]. In contrast, natural plant growth regulators derived from plants but not related to plant endogenous hormones are rarely reported. Therefore, the discovery and exploration of these novel natural plant growth regulators and analysis of their functions and the mechanisms underlying these functions will be important in improving our knowledge of plant growth regulators, reducing the pollution resulting from the manufacture of synthetic chemicals, and improving the yield and quality of crops.

Isopimaric acid is an abietane diterpene widely found in pine and cypress plants. It has various biological properties and has shown anticancer, antiviral, anti-inflammatory, and bacteriostatic activities [12,13]. A previous study extracted a significant quantity of isopimaric acid from *Thuja occidentalis* L., and subsequent activity assessment revealed the ability of this compound to regulate the growth of rice. However, the mechanism by which isopimeric acid regulates rice seedling growth remains unclear. To explore this mechanism, we conducted a statistical analysis on four phenotypic traits (shoot length, root length, shoot weight, and root weight) from 213 inbred lines of different rice lineages treated with 25 μg/mL isopimaric acid; then, using genome-wide association studies (GWAS), we integrated these four phenotypic traits with rice genetic data. We identified the candidate genes responsible for the regulation of rice growth by isopimaric acid. Further RT-qPCR analysis demonstrated that the relative expression levels of the resistance gene *OsASC1* and the antibody protein gene *OsBUD13* decreased significantly following treatment with 25 μg/mL isopimaric acid. These research results provide reference methods and a research basis for the discovery of new plant growth-regulating active substances and the study of their regulatory mechanisms.

## 2. Results

### 2.1. Isopimaric Acid Promotes the Growth of Nipponbare Rice

To screen the biological activity of isopimaric acid on rice, the Nipponbare seeds were treated with 25 μg/mL isopimaric acid or with 6.25, 12.5, 25, or 50 μg/mL GA_3_ for 15 days. The shoot length, shoot weight, root length, and root weight were measured and analyzed statistically. We found that treatment of the rice seeds with 6.25 μg/mL, 12.5 μg/mL, and 25 μg/mL GA_3_ increased shoot length by 61.69 ± 13.04%, 44.70 ± 13.26%, and 31.10 ± 9.33%, respectively, indicating that the lower the concentration of GA_3_, the more pronounced the increase (Figure 1A,C). However, the different concentrations of GA_3_ significantly decreased shoot weight, root length, and root weight (*p* < 0.05). After treatment with 6.25 μg/mL, 12.5 μg/mL, 25 μg/mL, and 50 μg/mL GA_3_, the root length decreased by 31.86 ± 10.53%, 40.96 ± 7.35%, 46.64 ± 8.11%, and 51.24 ± 5.54%, respectively (Figure 1C), while the root weight decreased by 44.79 ± 17.58%, 37.22 ± 15.29%, 34.51 ± 9.97%, and 36.96 ± 13.96%. The shoot weight decreased by 13.06 ± 11.38%, 16.81 ± 10.26%, and 30.81 ± 9.51% after treatment with 12.5 μg/mL, 25 μg/mL, and 50 μg/mL GA_3_, respectively (Figure 1B). The higher the concentration of GA_3_, the more obvious the inhibitory effect. However, treating the Nipponbare seeds with 25 μg/mL isopimaric acid significantly increased the resulting shoot length, root length, and shoot weight of the rice seedlings by 11.37 ± 5.05%, 12.96 ± 7.63%, and 27.98 ± 10.88%, respectively (Figure 1B,C) (*p* < 0.05).

### 2.2. Genome-Wide Association Studies in Rice Following Induction Treatment with Isopimaric Acid

To further investigate the mechanism by which isopimaric acid regulates rice growth and to identify the genes associated with these phenotypes, 213 inbred lines of different rice lineages were treated with 25 μg/mL isopimaric acid and allowed to grow, after which shoot length, shoot weight, root length, and root weight were measured. Subsequently, the GWAS was used to combine the phenotypic and genetic data to locate any associated candidate genes.

Statistical analyses of the phenotypic data showed that following induction with 25 μg/mL isopimaric acid, 15.02% of the rice varieties tested showed increased growth, 15.96% of the varieties showed decreased growth, and the growth of the remaining 69.02% did not show any significant change from the control. This indicated that isopimaric acid was able to regulate the growth of the rice seedlings, but this effect was not uniform across different rice varieties (Figure 2). In the rice varieties showing an increase in growth, the shoot length and shoot weight showed significant increases, accounting for 21.88% and 31.25% (Figure 2C,D). The root length and weight significantly increased, accounting for 6.25% and 3.13%, respectively (Figure 2A,B). These results suggest that treatment with isopimaric acid had a more obvious regulatory effect on the aboveground growth of rice seedlings than on the belowground growth.

Based on the GWAS analysis, the shoot lengths of rice seedlings were significantly associated with the 22,557,159 physical location interval on chromosome 2 (Figure 3C), while shoot weight was significantly associated with the 45,90,579 physical location interval on chromosome 8 (Figure 3D). No single nucleotide polymorphism (SNP) signals were detected as being associated with rice root length or weight (Figure 3A,B). Further analysis of the linkage disequilibrium blocks at these two intervals based on the Manhattan plot of shoot length and weight revealed that the SNP signals did not fall within the blocks. Therefore, the subsequent association analysis of candidate gene haplotypes with phenotypes was centered around the 600 kb range above the signal peak (Figure 3C,D). In this range, 98 annotated genes were found for shoot length, and 17 genes were significantly associated (including weak associations) with haplotypes. These included twelve genes encoding expressed proteins, two genes encoding hypothetical proteins, two genes encoding retrotransposon proteins, and one gene encoding a transposon protein (Table 1). Seventy-five annotated genes were found in this range for shoot weight, and four genes were significantly associated (including a weak association) with haplotypes, all of which expressed proteins (Table 2).

### 2.3. Haplotype Analysis and Tissue-Specific Expression of Candidate Genes

Gene annotation revealed that of the 17 candidate genes related to shoot length, *OsASC1* (*LOC_Os02g37080*) is an important resistance-related gene (Table 1). Based on the nonsynonymous SNPs detected in the exons, the gene can be divided into four haplotypes. The phenotypic values of Hap_1 and Hap_4 differed significantly (Figure 4A,B). In the analysis of tissue-specific expression of *OsASC1*, the most abundant expression was observed in the leaves (Figure 4C). The response to nutrient deprivation was the strongest response among the different abiotic stresses tested (Figure 4D).

Good annotation information was not available for any of the four candidate genes predicting shoot weight. However, OsBUD13 (LOC_Os08g08080) was better annotated than the others and was predicted to code an antibody (Table 2). Based on the non-synonymous SNPs detected in the exons, the gene was classified into two haplotypes, and the difference in the phenotypic values of Hap_1 and Hap_2 was significant (Figure 4E,F). In the analysis of tissue-specific expression of OsBUD13, the most abundant expression was observed in the leaves (Figure 4G).

### 2.4. Expression Analysis and Subcellular Localization of OSASC1 and OSBUD13

The expression of *OsASC1* and *OsBUD13* in the Nipponbare rice seedlings treated with 25 μg/mL isopimaric acid was investigated using RT-qPCR. The results showed that the expression of both *OsASC1* and *OsBUD13* was significantly lower than that of the control group (CK) (Figure 5A,B). Because *OsASC1* is an important resistance-related gene in rice, and *OsBUD13* is predicted to encode an antibody, these results suggest that isopimaric acid may regulate the growth of rice by affecting its defense mechanisms. To determine the localization of the proteins in plant cells, the transient expression of 35S::OsASC1-GFP and 35S::OsBUD13-GFP in *Nicotiana benthamiana* was used to determine their subcellular localization. The observation of protein expression was conducted using laser confocal microscopy. Fluorescent signals were observed in the cell membrane and nucleus for the control GFP, while OsASC1 was detected in the cell membrane and nuclear membrane and OsBUD13 was exclusively detected in the nucleus, indicating that OsBUD13 is a nuclear protein (Figure 5C).

## 3. Discussion

Phytohormones are organic signaling molecules produced by plants through their own metabolism, which can produce obvious physiological effects at low concentrations, and play important roles in plant growth, development, and environmental response processes [14]. To enhance crop yield and quality, a large number of synthetic commercialized plant growth regulators have been developed and are widely used in agricultural production. These include NAA, 2,4-dichlorophenoxyacetic acid (2,4-D), which has similar physiological effects to auxin and can promote cell elongation, inhibit the formation of adventitious roots, and prevent fruit set [15,16], and the cytokinin analog 6-BA, which can facilitate flower bud differentiation and improve root growth [9]. Gibberellins (GAs) are diterpenoid phytohormones that regulate plant growth and a wide range of developmental processes [17]. To date, more than 130 species of GAs have been characterized. GA_3_ is one of the most biologically active of these and is an efficient and broad-spectrum plant growth regulator [18,19]. Brassinosteroids (BRs) have the combined effects of gibberellin, cytokinin, and auxin [20,21,22].

In this study, we found that isopimaric acid can promote the growth of Nipponbare rice seedlings and shows similar activity to plant growth promoters. Exogenous application of 25 μg/mL isopimaric acid was more potent than GA_3_ in increasing the shoot weight, shoot length, and root length of seedlings. In addition, this concentration of isopimaric acid also showed different regulatory effects on different strains of rice. These results show that the diterpenoid isopimaric acid is a potential natural plant growth regulator. The regulation of plant growth and development by exogenous growth regulators usually involves complex endogenous hormone signaling responses and conduction pathways. These factors affect hormone receptors, positive and negative regulators, and downstream genes [23]. These molecules finely regulate hormone metabolism and activate or inhibit hormone signaling cascades, thereby regulating plant growth [24,25]. For example, after treatment with the exogenous methyl jasmonic acid (JA), the highly JA-responsive ERF109 mediates crosstalk between JA signaling and auxin biosynthesis, which can promote the formation of lateral roots in *Arabidopsis thaliana* by up-regulating the expression of two key enzyme-coding genes, i.e., *ASA1* and *YUC2*, involved in auxin biosynthesis [26]. Both exogenous auxins and cytokinins induce the expression of the cytokinin oxidase-coding gene, *OsCKX4*, in rice roots, whereas the auxin response factor OsARF25 can bind to the promoter of *OsCKX4*, activate its expression, and enhance cytokinin metabolism to promote root crown development in rice [27]. In addition to regulating plant growth and developmental processes, exogenous growth-regulating substances often affect plant resistance. In this study, GWAS revealed two candidate genes, the resistance-related gene, *OsASC1*, and an antibody-coding gene, *OsBUD13*. This suggests that isopimaric acid may affect the rice defense response.

Growth and defense responses are energy-consuming processes. Immune activation is typically accompanied by inhibition of growth, and when the defense response is inhibited, the energy used by the plant for growth and development will increase, indicating that there is a trade-off between defense responses and growth [28,29]. For example, in rice, PigmR enhances rice blast resistance but reduces yield [30]. JA is involved in a wide range of developmental processes, including root growth, stamen development, flowering time, and fruit ripening. Furthermore, as an important defense substance, JA plays a key role in regulating plant responses to various adversity-associated stresses [30,31,32]. It can activate the core transcription factor MYC2 and promote the expression of defense genes [33]. But when JA gradually accumulates, it will delay the degradation of repressors DELLAs by GAs and inhibit plant growth [34]. Under normal conditions, JA tends to inhibit plant growth and development while simultaneously increasing plant resistance [35,36]. In contrast to the above results, in this study, exogenous application of isopimaric acid significantly suppressed the expression of the defense-related gene *OsASC1*, *OsBUD13*, and promoted rice growth. Thus, isopimaric acid may promote plant growth by sacrificing some defenses. Enhanced nutrient uptake and competitiveness may therefore weaken plant defenses against biotic stress, reflecting the balance between growth and defense in plants.

## 4. Materials and Methods

### 4.1. Plant Materials and Growth Conditions

*Oryza sativa* L. ssp. japonica cv. Nipponbare and 213 inbred lines of different rice lineages (Rice Research Institute of Shenyang Agricultural University, Shenyang, China) were used for this experiment. Rice seeds were soaked in 3% (*v*/*v*) sodium hypochlorite solution for 15 min, washed with sterilized water (3–5 times), and then evenly spread into Petri dishes. The seeds were then cultivated at a constant temperature of 37 °C for 2–3 days. After germination, the Nipponbare seeds were transferred to Hoagland nutrient solution containing 25 μg/mL isopimaric acid or 6.25, 12.5, 25, or 50 μg/mL GA_3_. All treatments were performed with three replicates. The rice was then placed in the plant culture room at 25 °C, 16 h light/8 h dark, 1200 Lx (Illumination intensity), for 15 days. The Hoagland nutrient solution (Appendix A) was supplemented once every 3 d.

### 4.2. Genome-Wide Association Studies

The GWAS was performed using compressed mixed linear model (MLM) with four phenotypic traits (shoot length, root length, shoot weight, and root weight) from 213 inbred lines of different rice lineages treated with 25 μg/mL isopimaric acid. Significant SNPs sites were identified from their *p* values and R^2^, and *p* values were adjusted using the Bonferroni correction. The corrected *p* values were very conservative, and in the analysis, few marked rice genome association *p* values could be associated with genomic markers; therefore, the threshold was adjusted according to the conditions of the Manhattan plots. *p* < 1 × 10^−4^ was selected as the threshold for significant association between SNP markers and target traits, and the gene-related quantitative trait locis (QTLs) were then defined according to the rice population linkage disequilibrium (LD) attenuation distance of 600 kbp. SNPs with physical positions closer than 600 kbp on the same chromosome were considered the same QTL. If there were multiple significant SNPs within the LD range, those SNPs were considered one QTL, and the SNP with the smallest *p* value was considered the “lead” SNP. For SNPs with a physical distance greater than 300 kbp apart, QTL was divided by LD block in Haploview software (https://www.broadinstitute.org/haploview/haploview, accessed on 18 August 2024).

### 4.3. Candidate Gene Retrieval and Haplotype Analysis

The SNPs with the lowest *p* values in the colocating QTLs were used as sites, and all genes within a ± 300 kbp range of the SNPs were retrieved from the China Rice Data Center. Preliminary candidates were predicted using gene annotation, and haplotype analyses were then performed for all candidate genes within the range of lead SNPs ± 300 kbp using SNPs with nonsynonymous mutations. Candidate genes without significant differences in phenotypic values between haplotypes were then analyzed for haplotype using SNPs in the promoter region. Sequence comparison of each gene was determined using nonsynonymous SNPs associated with the trait, and the differences in phenotypic values between the haplotypes of each gene were calculated using one-way ANOVA or *t*-tests. If the results of one-way ANOVA were significant (*p* < 0.01), Duncan’s multiple range tests were performed for comparison.

### 4.4. Rna Extraction and Rt-Qpcr Analysis

Rice seedlings were taken from the control and experimental groups treated with 25 μg/mL isopimaric acid for 5 d. Total RNA was extracted using the TransZol Up RNA kit (Transgen Biotech, Beijing, China). First-strand cDNA synthesis was conducted using TransScript^®^ One-Step gDNA Removal and cDNA Synthesis SuperMix. The sequence information of the genes *OsASC1* and *OsBUD13* was obtained from China Rice Data Center (https://www.ricedata.cn/gene/, accessed on 18 August 2024). NCBI PrimerBlast was then used to design qPCR primers (Appendix A). SYBR Premix Ex Taq (Takara Bio., Dalian, China) and LightCycler (Roche, Mannheim, Germany) were used to conduct qPCR analysis, and the relative gene expression was then calculated using the 2^−ΔΔCT^ method.

### 4.5. Gene Cloning and Vector Construction

Amplification primers were designed with reference to *OsASC1* and *OsBUD13* gene sequences (Appendix A). The two genes were then amplified using PCR with Nipponbare rice seedling cDNA as templates (Appendix A). The coding sequences of *OsASC1* and *OsBUD13* were ligated into the pRI101-GFP vector using seamless cloning.

### 4.6. Subcellular Localization

Gene–GFP fusion expression vectors were then transformed into *Agrobacterium tumefaciens* strain GV3101 and grown on LB solid medium with Rif (50 mg/L) and Kan (50 mg/L) for 2 days. Single colonies were picked and cultured in liquid LB media containing the same antibiotics for activation. Then, 1 mL of activated bacteria was added to 10mL liquid Luria–Bertani (LB) media with Rif (50 mg/L) and Kan (50 mg/L) and then cultured at 28 °C, 200 rpm until the OD_600_ reached 0.8. The culture was then centrifuged at 8000 rpm for 5 min, and the same volume of buffer was subsequently used to re-suspend the bacterial cells. The suspension was left to stand for 2 h under dark conditions.

The transient expression experiments were conducted in four-week-old *Nicotiana benthamiana* plants. First, 1 mL bacterial suspension was injected into the dorsal part of each leaf. Then, the tobacco plants were incubated in the dark for 2 days, after which the dorsal injection area of the leaves was harvested and examined for a GFP fluorescence signal under an excitation wavelength of 488 nm using laser confocal microscopy.

### 4.7. Statistical Analysis

The data were expressed as mean ± standard deviations (SD). The data were analyzed by IBM SPSS Statistics 24.0. The *t*-test was used to compare data between 2 groups when the data followed a normal distribution. Statistical significance was assessed by one-way analysis of variance (ANOVA) followed by an LSD post hoc test. The results were considered statistically significant when *p* < 0.05. The data were analyzed using GraphPad Prism 8. The following formulas were used: (Control group-treatment group)/control group ∗ 100% to calculate the ratio of decrease and (Treatment group-control group)/treatment group ∗ 100% to calculate the ratio of increase.

## 5. Conclusions

Isopimaric acid is an isopimaricane diterpenoid found widely in conifers. Activity screening revealed that compared with the same concentration of GA_3,_ 25 μg/mL isopimaric acid has a better promotional effect on the weight and length of shoots and the root lengths of Nipponbare rice seedlings, indicating that isopimaric acid can be used as a natural active substance to regulate plant growth. Subsequently, we statistically analyzed four phenotypes in 213 inbred lines of different rice lineages treated with isopimaric acid and found that isopimaric acid may be more involved in the regulation of shoot length. The phenotypic data were integrated with rice genetic information using GWASs, revealing two relevant candidate genes, the resistance-related gene *OsASC1* and the antibody-coding gene *OsBUD13*. Subcellular localization demonstrated that OsASC1 was localized to the cell membrane and nuclear membrane, while OsBUD13 was localized in the nucleus. Further analysis using RT-qPCR revealed that the exogenous application of 25 μg/mL isopimaric acid significantly suppressed the expression of *OsASC1* and *OsBUD13*, suggesting that isopimaric acid may regulate seedling growth by affecting defense pathways in rice. In summary, this study provides a foundation for investigating active substances that regulate rice seedling growth and understanding their underlying regulatory mechanisms.

## Figures and Tables

**Figure 1 ijms-25-09161-f001:**
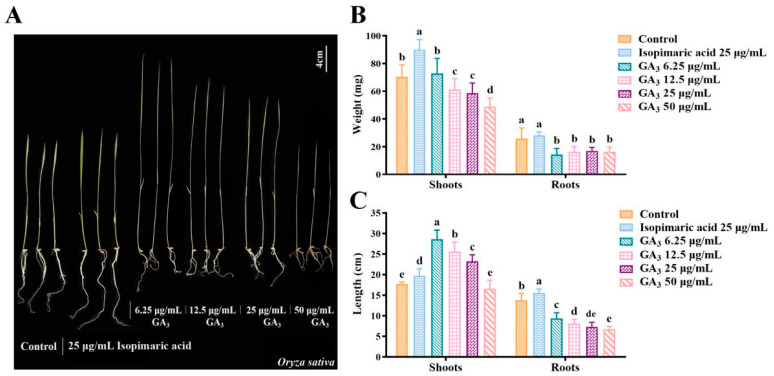
The effects of isopimaric acid on the growth of Nipponbare rice seedlings. (**A**) Rice growth 15 days following treatment with different concentrations of GA_3_ or 25 μg/mL isopimaric acid. (**B**,**C**) Effects of different concentrations of GA_3_ or 25 μg/mL isopimaric acid on seedling shoot length, root length, shoot weight, and root weight 15 days following treatment. Different letters indicate statistically significant differences (ANOVA, LSD, *p* < 0.05).

**Figure 2 ijms-25-09161-f002:**
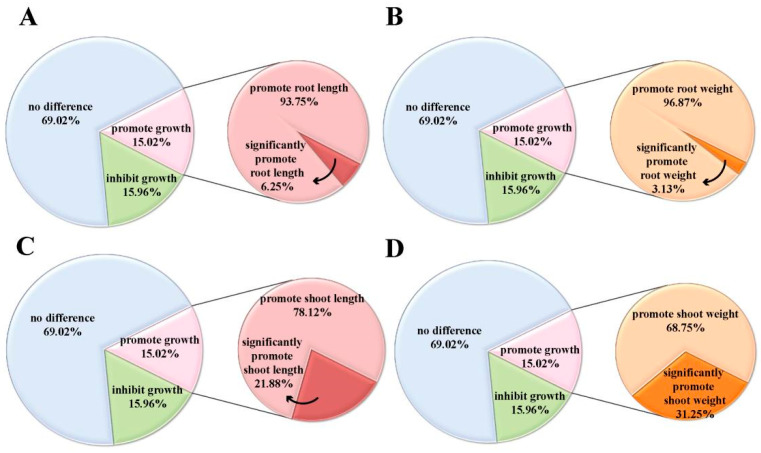
Phenotypes of different rice varieties following isopimaric acid treatment. (**A**,**B**) Root length and root weight phenotypes of 213 inbred lines of different rice lineages treated with 25 μg/mL isopimaric acid. (**C**,**D**) Shoot length and shoot weight phenotypes of 213 inbred lines of different rice lineages treated with 25 μg/mL isopimaric acid.

**Figure 3 ijms-25-09161-f003:**
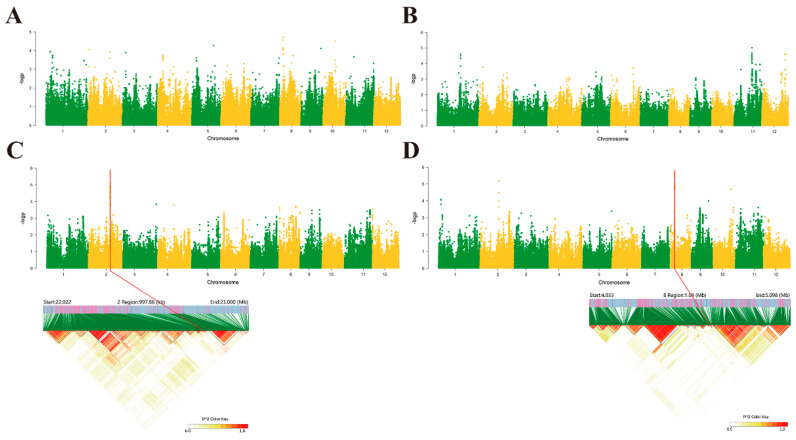
Manhattan plot and linkage disequilibrium blocks of four phenotypic and chromosomal associations in rice. (**A**–**D**) Manhattan plot of rice seedling root length, root weight, shoot length, and shoot weight. Linkage disequilibrium blocks of rice seedling shoot length and shoot weight.

**Figure 4 ijms-25-09161-f004:**
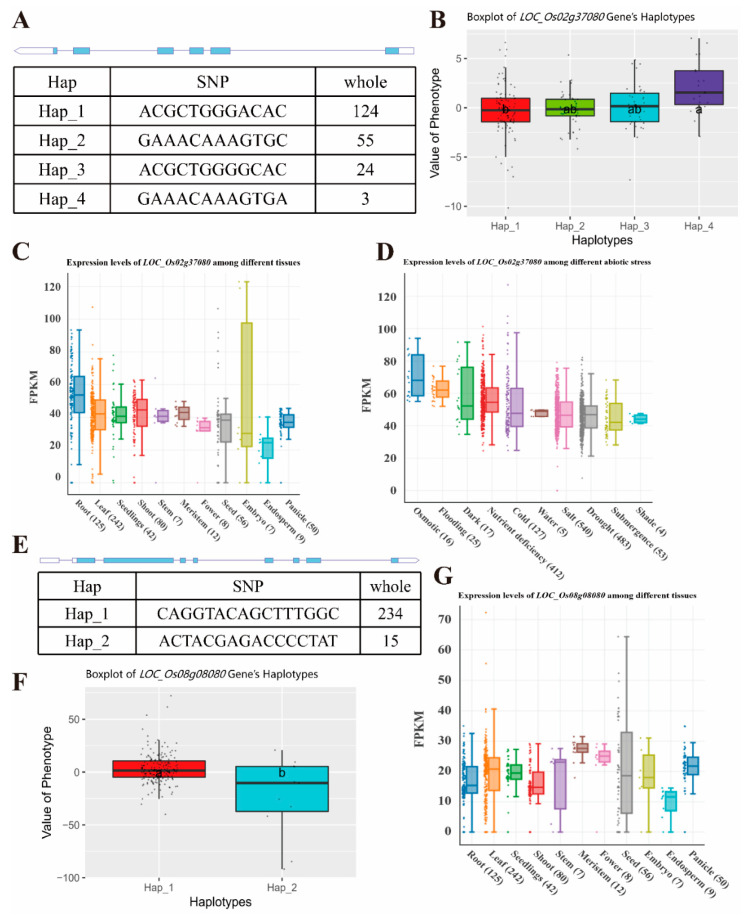
Haplotype analysis of *OsASC1* and *OsBUD13* and their tissue-specific expression. (**A**) The exon–intron structure of *OsASC1* and DNA polymorphisms in this gene. (**B**) Boxplot depicting the phenotypic values of the different haplotypes of *OsASC1*. (**C**) Expression of *OsASC1* in different rice tissues. (**D**) *OsASC1* expression under different abiotic stresses. (**E**) The exon–intron structure of *OsBUD13* and DNA polymorphisms in this gene. (**F**) Boxplot depicting the phenotypic values of the different haplotypes of *OsBUD13*. (**G**) Expression of *OsBUD13* in different rice tissues. Different lowercase letters (a and b) represent significant differences at the 0.05 level using one-way ANOVA with Tukey’s post hoc tests.

**Figure 5 ijms-25-09161-f005:**
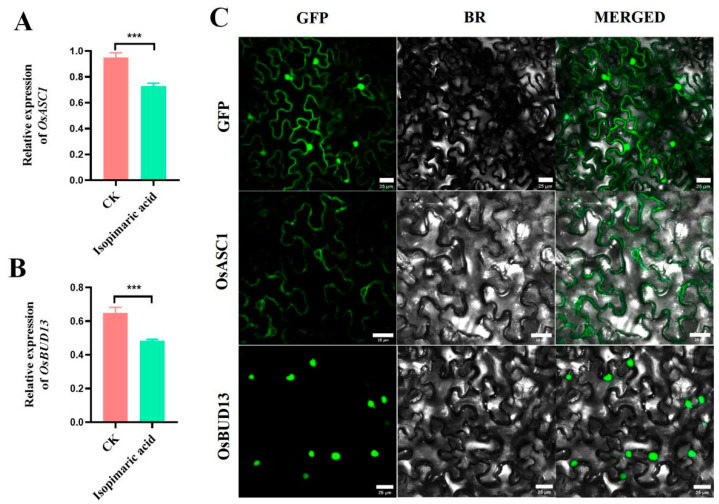
Expression analysis and subcellular localization of OsASC1 and OsBUD13. (**A**,**B**) qRT-PCR analysis of *OsASC1* and *OsBUD13* expression in CK (the control group) and rice seedlings treated with 25 μg/mL isopimaric acid (two-sample *t*-test, *** *p* < 0.001). (**C**) The subcellular localization of OsASC1 and OsBUD13. OsASC1-GFP and OsBUD13-GFP were expressed in *Nicotiana benthamiana*, and confocal microscopy revealed that OsASC1 was localized in both the cell membranes and nuclear membranes, while OsBUD13 was localized in the nucleus (scale bars, 25 μm). BR, bright field.

**Table 1 ijms-25-09161-t001:** Prediction of candidate genes of shoot length in rice.

Chr	Genes	Gene Name	Gene Position	Description
2	LOC_Os02g37080	Os02g0581300	22407730..22412305	ASC1, putative, expressed
2	LOC_Os02g37130	Os02g0582200	22445010..22448965	expressed protein
2	LOC_Os02g37250	Os02g0583700	22509550..22510620	hypothetical protein
2	LOC_Os02g37260	Os02g0584200	22529056..22531379	hypothetical protein
2	LOC_Os02g37309	Os02g0584900	22556916..22558121	expressed protein
2	LOC_Os02g37320	Os02g0585100	22558881..22559609	heavy metal-associated domain containing protein, expressed
2	LOC_Os02g37410	/	22595368..22596778	expressed protein
2	LOC_Os02g37430	Os02g0586500	22602418..22605754	LSM domain-containing protein, expressed
2	LOC_Os02g37440	Os02g0586600	22607190..22609208	expressed protein
2	LOC_Os02g37460	/	22614291..22615805	retrotransposon protein, putative, Ty3-gypsy subclass
2	LOC_Os02g37500	Os02g0587300	22636061..22636963	hypothetical protein
2	LOC_Os02g37540	Os02g0587800	22650959..22651932	virulence factor, pectin lyase fold family protein
2	LOC_Os02g37620	/	22702356..22705354	transposon protein, putative, CACTA, En/Spm sub-class
2	LOC_Os02g37654	Os02g0589000	22722799..22733189	lecithine cholesterol acyltransferase, putative, expressed
2	LOC_Os02g37750	Os02g0589900	22761941..22778592	lecithine cholesterol acyltransferase, putative, expressed
2	LOC_Os02g37760	Os02g0590000	22782750..22783121	diacylglycerol acyltransferase domain containing protein
2	LOC_Os02g37790	/	22797600..22798855	retrotransposon protein, putative, unclassified, expressed

**Table 2 ijms-25-09161-t002:** Prediction of candidate shoot weight genes in rice.

Chr	Genes	Gene Name	Gene Position	Description
8	LOC_Os08g07870	/	4424985..4429988	retrotransposon protein, putative, unclassified, expressed
8	LOC_Os08g07880	Os08g0176100	4430918..4434203	phosphopantothenate–cysteine ligase, putative, expressed
8	LOC_Os08g07930	Os08g0176633	4468741..4472698	resistance protein, putative, expressed
8	LOC_Os08g08080	Os08g0178300	4586478..4591217	BUD13, putative, expressed

## Data Availability

Data will be available upon request.

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
