# Peer review of "Genome-Wide Association Studies Reveal That the Abietane Diterpene Isopimaric Acid Promotes Rice Growth through Inhibition of Defense Pathways"

_ijms, 2024, doi:10.3390/ijms25179161_

Round 1

Reviewer 1 Report

Comments and Suggestions for Authors

Comments:

In this manuscript (ijms-3153042), a total of 213 inbred lines of different rice lineages were screened and found that isopimaric acid had different growth promotional activities on rice seedlings of different varieties. Combined with genome-wide association studies (GWAS), linkage disequilibrium block, and gene haplotype signiffcance analysis, SNP signals signiffcantly associated with the length and weight of shoots were identiffed on chromosomes 2 and 8, respectively. 17 candidate genes related to length of shoots and 4 candidate genes related to weight of shoots. Two genes related to this isopimaric acid regulation phenotype were identiffed as OsASC1 (LOC_Os02g37080) on chromosome 2 and OsBUD13 (LOC_Os08g08080) on chromosome 8. Here are some comments for this MS:

1.      Fig. 1 There are four treatments with different GA3 concentrations, and why there are only one concentration treatment of isopimaric acid ;

2.      Fig.2 The Left side pie charts are the same in A-D, and we suggest the authors reorganize the Fig.2;

3.      Fig.3 We suggest the authors to mark the peak Loci.

Author Response

Response to reviewer 1:

In this manuscript (ijms-3153042), a total of 213 inbred lines of different rice lineages were screened and found that isopimaric acid had different growth promotional activities on rice seedlings of different varieties. Combined with genome-wide association studies (GWAS), linkage disequilibrium block, and gene haplotype signiffcance analysis, SNP signals signiffcantly associated with the length and weight of shoots were identiffed on chromosomes 2 and 8, respectively. 17 candidate genes related to length of shoots and 4 candidate genes related to weight of shoots. Two genes related to this isopimaric acid regulation phenotype were identiffed as OsASC1 (LOC_Os02g37080) on chromosome 2 and OsBUD13 (LOC_Os08g08080) on chromosome 8. Here are some comments for this MS:

  1. Fig. 1 There are four treatments with different GA3 concentrations, and why there are only one concentration treatment of isopimaric acid ;

Response: In our previous study on the biological activity of isopimaric acid, 25, 50, and 100 μg/mL isopimaric acid were used to treat rice seedlings. The results showed that 25 μg/mL isopimaric acid had the most obvious promoting effect on shoot length, root length, shoot weight, and root weight (Wang, 2019). Therefore, the concentration of isopimaric acid applied in this study was all 25 μg/mL.

Wang, W.J. Study on Secondary Metabolites and Their Biological Functions of Thuja occidentalis (Chinese). Dissertation for Master, Shenyang Agricultural University, China, 2019.

  1. Fig.2 The Left side pie charts are the same in A-D, and we suggest the authors reorganize the Fig.2;

Response: Because there are many parameters, after our thinking, we felt that merging could not clearly show our data results, so we did not modify this figure.

  1. 3 We suggest the authors to mark the peak Loci.

Response: We modified Fig.3 to correlate the manhattan plot and linkage disequilibrium blocks.

Reviewer 2 Report

Comments and Suggestions for Authors

Dear Editor

Many thanks for considering me as a potential reviewer for the article "Genome-wide association studies reveal that the abietane diterpene isopimaric acid promotes rice growth through inhibition of defense pathways". No doubt, the article is well-structured, presented and well written. However, I have some observations, which I believe should be taken into consideration before proceeding further.

Comments

1. Generally, the aims and objectives section at the end of the introduction must be revised, the article is poorly cited, and the discussion section must be improved.

2.      Add quantitative results in the abstract and the concluding remarks at the end of the Abstract section should be revised.

3.      Introduction: please cite this information “Depending on their function, plant growth regulators can be divided into plant growth promoters, plant growth retardants, or plant growth inhibitors. Of these, plant growth promoters are the most versatile and commonly utilized in agriculture.”

4.      The authors revealed that isopimaric acid can inhibit defense pathways to promote rice seedling growth, can you explain the potential molecular mechanism and how OsBUD13 and OsASC1 contribute in this mechanism?

5.      Majority of plants e.g. Nicotiana benthamiana lack authority names, please write authority names, accordingly, throughout manuscript,

6.      Can you justify or add some information on the large-scale agricultural practices of this mechanism?

7.      The section for statistical analysis is missing in the manuscript (suggested to add).

8.      How did the authors find the percentage increase and/or decrease as compared to control, please mention the formula under the sub-section statistical analysis (suggested to be added), and make it reader-friendly.

9.      Figure 1 B and C, I think there will be no need to mention the same treatment level in the legends, just once will be enough.

10.  Section 4.6, Please use LB solid medium, since LB appears the first time, and please write its full form.

11.  Section 2.4, OsBUD13 was significantly lower than that of the control group (Figure 5A, B), you mention in the text control, while in the graph it’s CK, either use same expression and either write it in the graph description, avoid confusion.

12.  Discussion section; please make this sentence shorter and cite ‘The regulation of plant growth and development by exogenous growth regulators usually involves complex endogenous hormone signaling responses, conduction pathways, and the interaction of multiple hormone pathways, which affect hormone receptors, positive and negative regulators, and the downstream genes’.

13.  Page 9, please compare your results with previous reports ‘In this study, we found that isopimaric acid, an abietane diterpene derived from pine and cypress plants, can promote the growth of Nipponbare rice seedlings, and 25 μg/mL isopimaric acid was more potent than GA3 in increasing the shoot weight, shoot length, and root length of seedlings. In addition, this concentration of isopimaric acid also showed different regulatory effects on different strains of rice. These results show that the diterpenoid isopimaric acid is a potential natural plant growth regulator.

14.  Page 9, please compare your results with available literature ‘In this study, GWAS revealed two candidate genes, the resistance-related gene OsASC1, and an antibody-coding gene OsBUD13. Exogenous application of 25 μg/mL isopimaric acid significantly suppressed the expression of both OsASC1 and OsBUD13. Thus, isopimaric acid may promote plant growth by sacrificing some of the defenses. Enhanced nutrient uptake and competitiveness may therefore weaken plant defenses against biotic stress, reflecting the balance between growth and defense in plants.

Author Response

Response to reviewer 2:

Many thanks for considering me as a potential reviewer for the article "Genome-wide association studies reveal that the abietane diterpene isopimaric acid promotes rice growth through inhibition of defense pathways". No doubt, the article is well-structured, presented and well written. However, I have some observations, which I believe should be taken into consideration before proceeding further.

Comments

  1. Generally, the aims and objectives section at the end of the introduction must be revised, the article is poorly cited, and the discussion section must be improved.

Response: The aims and objectives section at the end of the introduction was revised. “These research results provide reference methods and research basis for the discovery of new plant growth regulating active substances and the study of their regulatory mechanisms.” .

  1. Add quantitative results in the abstract and the concluding remarks at the end of the Abstract section should be revised.

Response: We have added the following quantitative results in the abstract. “We used activity screening to demonstrate that treatment of the Nipponbare seeds with 25 μg/mL isopimaric acid significantly increased resulting shoot length, root length, and shoot weight of rice seedlings by 11.37 ± 5.05 %, 12.96 ± 7.63 %, and 27.98 ± 10.88 %, and that it has a higher activity than does GA3 at the same concentration. A total of 213 inbred lines of different rice lineages were screened and we found that isopimaric acid had different growth promotional activities on rice seedlings of different varieties. In induction with 25 μg/mL isopimaric acid, 15.02 % rice varieties tested showed increased growth, while 15.96 % of the varieties showed decreased growth, and the growth of the remaining 69.02 % did not show any significant change from the control. In the rice varieties showing an increase in growth, the shoot length and shoot weight significant increased accounted for 21.88 % and 31.25 %. The root length and weight significantly increased accounted for 6.25 % and 3.13 %.”

  1. Introduction: please cite this information “Depending on their function, plant growth regulators can be divided into plant growth promoters, plant growth retardants, or plant growth inhibitors.Of these, plant growth promoters are the most versatile and commonly utilized in agriculture.”

Response: “Depending on their function, plant growth regulators can be divided into plant growth promoters, plant growth retardants, or plant growth inhibitors. Of these, plant growth promoters are the most versatile and commonly utilized in agriculture (Farjaminezhad et al., 2019).” (DOI: 10.1007/s13205-019-1836-z)

  1. The authors revealed that isopimaric acid can inhibit defense pathways to promote rice seedling growth, can you explain the potential molecular mechanism and how OsBUD13 and OsASC1 contribute in this mechanism?

Response: There is no research progress has been reported on these two genes in rice or other grasses. Therefore, based on the fundamental annotation information and quantitative detection results from our study, we hypothesize that isopimaric acid may regulate rice growth by influencing its defense mechanisms. However, the specific molecular mechanisms, such as the synergistic regulation of growth and defense related genes, which subsequently affect the levels of growth and defense related hormones, ultimately leading to phenotypic changes still require further investigation.

  1. Majority of plants e.g. Nicotiana benthamianalack authority names, please write authority names, accordingly, throughout manuscript. 

Response: We modified the name of rice in the method to Oryza sativa L. ssp. japonica cv. Nipponbare. In the relevant literature reviewed, the tobacco names used in the transient transformation experiments were written as Nicotiana benthamiana. Therefore, we have not changed the names used in this article.

  1. Can you justify or add some information on the large-scale agricultural practices of this mechanism?

Response: At present, most studies on the balance between growth and defense in plants focus on the molecular level, and the application in agricultural production is relatively lacking. Therefore this manuscript does not describe it.

  1. The section for statistical analysis is missing in the manuscript (suggested to add).

Response: The statistical analysis is supplemented in the methods. “Data were expressed as mean ± standard deviations (SD). Data were analyzed by IBM SPSS Statistics 24.0. The t-test was used to compare data between 2 groups when data followed a normal distribution. Statistical significance was assessed by one-way analysis of variance (ANOVA) followed by LSD post hoc test. Results were considered statistically significant when P < 0.05. Data were analyzed using GraphPad Prism 8. (Control group-treatment group)/control group *100% to calculate the ratio of decrease. (Treatment group-control group)/treatment group*100% to calculate the ratio of increase.”

  1. How did the authors find the percentage increase and/or decrease as compared to control, please mention the formula under the sub-section statistical analysis (suggested to be added), and make it reader-friendly.

Response: Referring to the comments 7, the calculation method is added to the statistical analysis section.

  1. Figure 1 B and C, I think there will be no need to mention the same treatment level in the legends, just once will be enough.

Response: Figure B shows the effects of different concentrations of GA3 or 25 μg/mL isopimaric acid on seedling shoot weight, and root weight. Figure C shows the effects on shoot length and root length. We think it is still need to annotate the figures separately.

  1. Section 4.6, Please use LB solid medium, since LB appears the first time, and please write its full form.

Response: Have been revied.

  1. Section 2.4, OsBUD13was significantly lower than that of the control group (Figure 5A, B), you mention in the text control, while in the graph it’s CK, either use same expression and either write it in the graph description, avoid confusion.

Response: We write additional explanations in the text and graph description. “The results showed that the expression of both OsASC1 and OsBUD13 was significantly lower than that of the control group (CK) (Figure 5A, B).” “Expression analysis and subcellular localization of OsASC1 and OsBUD13. (A, B) qRT-PCR analysis of OsASC1 and OsBUD13 expression in CK (the control group) and rice seedlings treated with 25 μg/mL isopimaric acid.”

  1. Discussion section; please make this sentence shorter and cite ‘The regulation of plant growth and development by exogenous growth regulators usually involves complex endogenous hormone signaling responses, conduction pathways, and the interaction of multiple hormone pathways, which affect hormone receptors, positive and negative regulators, and the downstream genes’.

Response: We modified this sentence and added citation. “The regulation of plant growth and development by exogenous growth regulators usually involves complex endogenous hormone signaling responses and conduction pathways. These factors affect hormone receptors, positive and negative regulators, and the downstream genes (Roychoudhury et al., 2021).” DOI: 10.1007/s00299-021-02755-9

  1. Page 9, please compare your results with previous reports ‘In this study, we found that isopimaric acid, an abietane diterpene derived from pine and cypress plants, can promote the growth of Nipponbare rice seedlings, and 25 μg/mL isopimaric acid was more potent than GA3in increasing the shoot weight, shoot length, and root length of seedlings. In addition, this concentration of isopimaric acid also showed different regulatory effects on different strains of rice. These results show that the diterpenoid isopimaric acid is a potential natural plant growth regulator.

Response: There are not many studies on the activity of isopimaric acid on plant growth. We did not find good literature, but I changed the description of this paragraph. “In this study, we found that isopimaric acid can promote the growth of Nipponbare rice seedlings, and shows similar activity to plant growth promoters. And exogenous application of 25 μg/mL isopimaric acid was more potent than GA3 in increasing the shoot weight, shoot length, and root length of seedlings. In addition, this concentration of isopimaric acid also showed different regulatory effects on different strains of rice. These results show that the diterpenoid isopimaric acid is a potential natural plant growth regulator.”

  1. Page 9, please compare your results with available literature ‘In this study, GWAS revealed two candidate genes, the resistance-related gene OsASC1, and an antibody-coding gene OsBUD13. Exogenous application of 25 μg/mL isopimaric acid significantly suppressed the expression of both OsASC1 and OsBUD13. Thus, isopimaric acid may promote plant growth by sacrificing some of the defenses. Enhanced nutrient uptake and competitiveness may therefore weaken plant defenses against biotic stress, reflecting the balance between growth and defense in plants.

Response: “It can activate the core transcription factor MYC2 and promote the expression of  defense genes (Wang et al., 2019). But when JA gradually accumulates, it will delay the degradation of repressors DELLAs by GAs and inhibit plant growth (Yang et al., 2012). Under normal conditions, JA tends to inhibit plant growth and development, while simultaneously increasing plant resistance. In contrast to the above results, in this study, exogenous application of isopimaric acid significantly suppressed the expression of defense related genes gene OsASC1, OsBUD13, and promoted rice growth. Thus, isopimaric acid may promote plant growth by sacrificing some of the defenses. Enhanced nutrient uptake and competitiveness may therefore weaken plant defenses against biotic stress, reflecting the balance between growth and defense in plants.” DOI: 10.1038/s41477-019-0441-9; DOI: 10.1073/pnas.1201616109.